# Study of the Technological Properties of *Pedrosillano* Chickpea Aquafaba and Its Application in the Production of Egg-Free Baked Meringues

**DOI:** 10.3390/foods12040902

**Published:** 2023-02-20

**Authors:** Paula Fuentes Choya, Patricia Combarros-Fuertes, Daniel Abarquero Camino, Erica Renes Bañuelos, Bernardo Prieto Gutiérrez, María Eugenia Tornadijo Rodríguez, José María Fresno Baro

**Affiliations:** Lactic Acid Bacteria and Technological Application Group (BALAT), Food Hygiene and Technology Department, Faculty of Veterinary, University of León, 24071 León, Spain

**Keywords:** *Pedrosillano* chickpea, aquafaba, chemical composition, functional properties, egg-free meringues, sensory characteristics

## Abstract

Aquafaba is a by-product derived from legume processing. The aim of this study was to assess the compositional differences and the culinary properties of *Pedrosillano* chickpea aquafaba prepared with different cooking liquids (water, vegetable broth, meat broth and the covering liquid of canned chickpeas) and to evaluate the sensory characteristics of French-baked meringues made with the different aquafaba samples, using egg white as a control. The content of total solids, protein, fat, ash and carbohydrates of the aquafaba samples were quantified. Foaming and emulsifying capacities, as well as the foam and emulsions stabilities were determined. Instrumental and panel-tester analyses were accomplished to evaluate the sensory characteristics of French-baked meringues. The ingredients added to the cooking liquid and the intensity of the heat treatment affected the aquafaba composition and culinary properties. All types of aquafaba showed good foaming properties and intermediate emulsifying capacities; however, the commercial canned chickpea’s aquafaba was the most similar to egg white. The aquafaba meringues showed less alveoli, greater hardness and fracturability and minimal color changes after baking compared with egg white meringues; the meat and vegetable broth’s aquafaba meringues were the lowest rated by the panel-tester and those prepared with canned aquafaba were the highest scored in the sensory analysis.

## 1. Introduction

Legumes are a group of edible seeds from the *Leguminoseae* family, which include, among others, beans, chickpeas, peas, lentils or soybeans. According to the FAO [1], the production of legumes has increased 25% between 2010 and 2020 in the world; however, their consumption continues to be scarce. This low intake is associated with several factors: social changes and current lifestyle (people eating more outside of the home) [2]; very long preparation times (long soaking and cooking times) [3]; or difficulty for some people to digest them, causing belly swelling due to the production of large amounts of gases [4]. However, in recent years, consumers are demanding more and more plant origin foods in their diet instead of animal origin foods, mainly for environmental sustainability, animal health and animal welfare reasons [5,6], which may contribute to increasing their intake, either in traditional recipes or in the form of flour, processed purees, creams and prepared and/or canned legumes.

One of the typical legumes of Spanish Gastronomy are chickpeas (*Cicer arietinum*) and Castilla y León; Castilla-La Mancha and Andalucia are the regions with the largest area of cultivation [7]. In Spain, one of the most appreciated varieties, both in traditional and in avant-garde cuisine (stews, salads, broths or stir-fries), is the *Pedrosillano* chickpea (a variety of small size and almost spherical-shaped chickpeas, which are characterized by their small beak and a yellow-orange coloration and a firm structure that acquires a buttery texture after cooking).

The use of products or by-products derived from the processing of legumes would be a further step in the improvement of sustainable food production. One of the most important by-products generated during the legume processing is the aquafaba. Aquafaba is defined as the liquid obtained after cooking legumes, which includes both that obtained in traditional cooking (in water or broths) and the covering liquid of canned legumes [8]. The presence of some proteins, complex carbohydrates and other flavour components turns aquafaba into an ingredient of exceptional culinary quality due to its functional properties (foaming, emulsifying, binding, gelling or thickening) [9]. In fact, the use of aquafaba in the formulation of new vegan products, both sweet and savoury, such as foams, emulsions or dairy substitutes, has increased nowadays [10].

In most of the vegan products, aquafaba is used as an egg substitute, not only by sharing several technological properties but also because the production of legumes causes a lower environmental impact than the production of eggs or another animal protein sources [11]. Studies carried out by Nette et al. [12] reported that products made with legumes generate 35% less greenhouse gases than those made with eggs. However, it is necessary to consider that a more recent study [13] questioned the environmental benefits of aquafaba as an egg substitute in food processing, highlighting that it could have a negative impact on the environment footprint as a result of higher electricity requirements. In addition, the use of aquafaba would also be compatible with the concept of upcycling, which refers to the process of finding a new use for by-products [14] that could become innovative culinary options.

The use of aquafaba as an alternative to eggs and dairy products not only satisfies the needs of vegan people, but also of a growing number of consumers who prefer dietary ingredients rich in fibre and/or are allergen-free [5,10]. Egg is one of the 14 food allergens with mandatory declaration in the list of ingredients [15]. The egg white, particularly the ovomucoid, is more allergenic than the yolk and is problematic for allergic people in the preparation of those recipes and elaborations in which it is the main ingredient [16]. In this sense, the aquafaba has garnered attention in the food industry for a new generation of natural substitutes and egg- and dairy-free labels, without the need for protein isolates, edible gums or modified starches [10,17,18,19].

The foaming and stabilizing capacity of the classical meringues are defined by the egg proteins due to its amphiphilic nature [20]. In the aquafaba, the foam production is also possible due to the presence of the chickpeas’ proteins. Those proteins require a partial denaturation so that the hydrophobic amino acids were oriented towards the air bubbles and the hydrophilic ones towards the aqueous phase favouring the maintenance of the structure [21,22]. In the preparation of aquafaba, it is necessary that enough chickpea proteins are dissolved in the cooking broth so that when the air was mechanically introduced, the foams were generated and stable [23]. Other factors, such as the content of intrinsic carbohydrates (especially polysaccharides), the pH of the medium or the presence of fat in the aquafaba, could play a very important role in the foam formation and stability [9].

In recent years, some works on the foaming and emulsifying properties of aquafaba [23,24,25,26,27,28,29], the application of ultrasound technology in the extraction [30], the optimization of cooking conditions and pH [31] and the impact of concentration methods on the characteristics of spray-dried powders of soybean aquafaba [32] were published. Most of these studies were focused on the use of aquafaba in the preparation of vegan mayonnaise with limited studies on its chemical composition or its applications in other culinary elaborations, such as meringues or sponge cakes [23,33]. In general, the studies on aquafaba have been based on the use of aquafaba derived from commercial canned legumes; to the best of the authors’ knowledge, the studies of the chemical characteristics and functional properties of the aquafaba obtained during the preparation of culinary dishes and their impact on the functional and sensory properties are very limited or have not been addressed [34].

From the scientific point of view, the original hypothesis of this research work was: “could the cooking liquid of *Pedrosillano* chickpeas under different culinary recipes, widely established in gastronomy, be a good substitute for eggs as an ingredient in the manufacture of sweet or salty baked meringues?”.

In order to test this hypothesis, the aim of this study was to assess the compositional differences, as well as the foaming and the emulsifying properties of *Pedrosillano* chickpea aquafaba, prepared with different cooking liquids (water, vegetable broth, meat broth and the covering liquid of canned chickpeas) and to evaluate the sensory characteristics of French-type baked meringues made with the different aquafaba samples using egg white as a control.

## 2. Materials and Methods

### 2.1. Preparation the Aqueous Broths for Cooking Chickpeas

The different aqueous broths used to cook the chickpeas were prepared as follows

(1) Water: 4 g salt were dissolved in 3 L distilled water.

(2) Vegetable broth: prepared with 210 g cabbage, 105 g turnip, 175 g onion, 160 g leek, 260 g carrot, 20 g garlic and 4 g salt. All the ingredients were washed, drained, chopped, weighed and, finally, cooked with 3 L of distilled water for 40 min in a pressure cooker at 100 kPa (Excellent BRA, Valls, Spain). Next, the broth was filtered through a cotton cloth and stored refrigerated (4 °C) for a maximum of 24 h until it was used to cook the chickpeas.

(3) Meat broth: was prepared using the same ingredients and amounts used for the vegetable broth and 97 g fresh chorizo, 204 g marinated and cured pork ribs, 222 g beef shank, 219 g chicken, 153 g fresh salted pork ear and 210 g ham bone (these last two products were previously desalted in water for 12 h). All the ingredients were introduced into a pressure cooker (Excellent BRA, Spain) with 3 L of distilled water where they were cooked for 40 min at 100 kPa. Then, the broth was filtered, kept refrigerated (4 °C) for 24 h and defatted before being used to cook the chickpeas.

### 2.2. Cooking Conditions of Chickpea to Obtain Aquafaba

*Pedrosillano* chickpeas (Polifer S.L., La Bañeza, León, Spain) were used in all the experiments. Hydration was carried out at room temperature by introducing 300 g chickpeas into a container with 2 L distilled water and 6 g salt for 12 h. Then, the chickpeas were drained, washed and weighed.

Each broth described in the Section 2.1 (except for the canned chickpea aquafaba) was used to cook the chickpeas for 35 min in a pressure cooker (Excellent BRA, Spain) at 100 kPa using a ratio of chickpea–broth of 1:2 (*w*/*v*). Two replicates were carried out for each elaboration. The different types of aquafaba were obtained by draining the chickpeas.

The canned chickpea aquafaba was obtained by draining the covering liquid from 9 jars of 570 g commercial canned *Pedrosillano* chickpeas (Legumbres Penelas S.L., Villarejo del Órbigo, León, Spain).

All types of aquafaba were frozen at −30 °C in jars of approximately 100 g until analysis.

### 2.3. Analysis of the Chemical Composition of Aquafaba

The chemical composition of the different types of aquafaba was determined in duplicate following the methods described by Stantiall et al. [25]. Total solids content was analysed by drying the samples in an oven at 100 °C until constant weight. Proteins were quantified following the Kjeldahl method using an automatic distiller (Digestor Kjeldahl Tecator 1002, FOSS IBERIA, S.A, Barcelona, Spain); a value of 6.25 was used as the conversion factor [35]. The fat content was determined following the AOAC standard 920.39 method using a Soxhlet extraction system [36]. The ash content was quantified by gravimetry [37] using a dry ashing method with a muffle at 500 °C for 6 h. Finally, the quantification of carbohydrates was carried out by subtracting the content of protein, fat and ash from total solids content.

### 2.4. Analysis of Foaming and Emulsifying Properties of Aquafaba and Egg White

The foaming properties were determined in duplicate following the method described by Stantiall et al. [25]. A total of 80 mL of aquafaba or egg white (as control) was taken in a glass and mixed with a hand blender (Miniquick 5 MQ5000, Braun, Neu-Isenburg, Germany) for 4 min. Next, the content was transferred to a 250 mL cylinder and the volume of the foam generated at zero time and after 40 min was measured. The foaming capacity (FC) and the foam stability (FS) were calculated using the following equations and expressed as the percentage of overrun, which is defined as the amount of air incorporated into the liquid phase.
FC (%) = ((V_Final_ − V_Initial_)/V_Initial_) × 100 (1)
where V_Final_ is the volume of the foam generated after homogenization and V_Initial_ is the volume of aquafaba or egg white before homogenization.
FS (%) = ((V_40_ − V_Initial_)/V_Initial_) × 100 (2)
where V_40_ is the volume of the foam after 40 min and V_Initial_ is the initial volume of the foam generated after homogenization.

The emulsifying properties were determined in duplicate by spectrophotometry following the protocol described by Karaca et al. [38]. To compare the emulsifying capacity of aquafaba and egg white, 10 mL of each sample was mixed with 10 mL of olive oil using an Ultraturrax homogenizer (Unidrive 1000D Ingenieurbüro CAT, M.Zipperer GmbH, Ballrechten-Dottingen, Germany) for 1 min at 3000 rpm. A total of 5 mL of a 1% sodium dodecyl sulfate (SDS) solution was placed in a beaker, then 50 µL of the emulsion was added. Then the mixture was homogenized, and the absorbance was measured at 500 nm for 40 min taking the absorbance values at 10 min intervals. The emulsifying capacity (EC) and the emulsion stability (ES) were calculated using the following equations:EC (m^2^⁄g) = ((2 × 2.303 × A_0_)/% Protein) (3)
where A_0_ is the absorbance of the diluted emulsion immediately after homogenization and (% Protein) is the weight of protein per volume (g/mL).
ES (min) = A_0_/ΔA × t (4)
where A_0_ is the absorbance of the diluted emulsion immediately after homogenization, ΔA is the change in absorbance between 0 and 10 min (A_0_–A_10_) and t is the time interval (10 min).

### 2.5. Baked French Meringue Preparation

The culinary applicability of the different types of aquafaba compared with egg white was assessed by preparing, in duplicate, different batches of French meringues using the following recipe: 100 g aquafaba or egg white, 100 g icing sugar and 1 mL natural lemon juice. All the ingredients were mixed with a hand blender (Miniquick 5 MQ5000, Braun, Neu-Isenburg, Germany) for 1 min at medium speed and, later, at maximum power until they were stiff. The meringue mixture was deposited on a *silpat* with the help of metal moulds to obtain meringues of 5 cm in diameter and 1 cm of height. Next, the meringues were baked in an oven (Conterm, Selecta, S.A., Barcelona, Spain) at 120 °C for 1.5 h, cooled to room temperature and stored in hermetically closed containers to protect them from the moisture until their analysis.

### 2.6. Sensory Analysis of Meringues

The colour analysis of the meringues was accomplished using a Konica CM reflectance colourimeter (Minolta, Osaka, Japan). The MAV (measurement/illumination area) and MAV mask pattern were read with 8 mm diameter glass. Data was processed using the program Color Data Software CM-S100w SpectraMagic TM NX ve. 1.9, Pro USB (Konica, Minolta, Osaka, Japan). Nine measurements on the surface of each meringue were carried out for each type of meringue and each batch elaborated. In all colour determinations, the equipment was previously calibrated for zero and white using standard plates (illuminate D65 and the 10° SCI observer). The colour parameters of the CIE*Lab* scale were studied: lightness (*L**), red-green component (*a**) and yellow-blue component (*b**).

The instrumental analysis of the meringue’s texture was performed following the method proposed by Stantiall et al. [25] with some modifications. A TA.XT2i texturometer and the Texture Expert program, v1.20 (Stable Micro Systems, Godalming, Surrey, UK) were used. The assays were carried out with a p 40 probe at a constant speed of 0.5 mm/s and using a degree of compression of 50%. Fracturability, brittleness, hardness and elasticity were quantified. In each type of meringue and each batch, 8 determinations of the texture profile were carried out.

The proportion of gas cells in the meringues (alveoli), expressed as the percentage of porosity, was determined by taking cross section photographs of the meringues, which were subsequently divided into portions of 1 cm length. The images were processed with the program Gimp 2.10.10 (GNOME, 2019) using the histogram tool. The alveoli percentage was obtained through the percentage of certain colour strip pixels [39]. For each meringue batch, the alveoli analyses were carried out in four meringues and three areas of each meringue.

Finally, the analysis of four sensory attributes (appearance, smell, taste and texture) was carried out using a seven-point hedonic scale for each attribute (1 = dislike extremely; 2 = dislike very much; 3 = dislike somewhat; 4 = neither like nor dislike; 5 = like somewhat; 6 = like a lot; and 7 = like very much) [40]. Likewise, a global impression score was accomplished using a scale from 1 (very bad) to 10 (very good). These analyses were performed by a panel of 114 untrained tasters.

### 2.7. Statistical Analysis

The statistical analysis was performed using the software SPSS statistics v.26.0 [41]. All variables were tested for the assumptions of normality and homoscedasticity. An analysis of variance (ANOVA) was applied to study the differences on the chemical composition of the different types of aquafaba and the differences on the sensory parameters of the meringues (data sets with normal distribution); subsequently, in those cases in which significant differences were observed, the Tukey test was applied to compare differences among groups. The Kruskal–Wallis test was used to study the differences on the foaming and emulsifying properties of the samples (data sets without normal distribution); in the cases in which significant differences were observed the U of the Mann–Whitney test was performed to define the differences among groups. A *p* < 0.05 was considered to be significant.

## 3. Results and Discussion

### 3.1. Chemical Composition of Aquafaba

The results of the chemical composition analysis of the different types of aquafaba were collected in Table 1.

The water aquafaba as well as the vegetable broth aquafaba showed a very similar total solids, protein and carbohydrate contents, with values ranging 5.63–5.84, 1.19–1.21 and 3.36–3.79 g/100 g aquafaba, respectively. On the contrary, the ash contents showed significant differences (*p* < 0.05) between them. The vegetable broth aquafaba had an ash content approximately 33% higher than water aquafaba due to the presence of minerals from the vegetables. Neither of the aquafaba samples contained fat in its composition.

The results of total solids, protein, ash and fat contents in the water aquafaba were very similar to those described by Bird et al. [42], Stantiall et al. [25] and He et al. [26]. The minor variations observed among the studies would be associated with differences in the soaking and cooking conditions of the chickpeas (time or water–chickpea ratio) and also related to the chickpea variety [9].

The meat broth aquafaba contained the highest total solids value due to the elevated protein and fat contents, which together represented almost 60% of the total solids. An important part of the proteins come from the meat sarcoplasm and connective tissue that were solubilized during the cooking. Furthermore, although the meat broth was defatted before being used to cook the chickpeas, the resulting aquafaba showed the highest fat content. The difficulty to eliminate the fat of the meat broth is probably related to its incorporation as an emulsion, stabilized by the solubilized proteins after the cooking process, in the aqueous phase. The ash content was similar to that reported for the vegetable broth aquafaba, but its carbohydrate concentration was a 35% lower with a value of 2.20 g/100 g aquafaba.

The canned and the meat broth aquafaba showed similar ash and protein contents; however, significant differences (*p* < 0.05) on their carbohydrates and fat contents were observed. The proteins represented approximately 40% of total solids in the canned chickpeas aquafaba and 30% in the meat broth aquafaba. The total solids content of the commercial aquafaba used in this study was very similar to that described by Shim et al. [34]; however, it showed between a 45 and 90% higher protein content, about 2.6 times more ash content and 50% less carbohydrates, than those described by other authors [27,29,34]. These differences might be associated to the chickpea variety; He et al. [26] reported important variations in the chemical composition and the properties of aquafaba samples obtained from different types of chickpeas. Likewise, since there is a relationship between the proportion of broth and the concentration of the extracted compounds [43], these differences may be also associated with the chickpea–water ratio used in the preparation. In the same way, another important factor that may affect the chemical composition of the aquafaba would be the variations in the cooking conditions. High temperatures combined with prolonged cooking times would lead to significant changes in the seed coat of the chickpeas, which allows the passage of a greater or lesser amount of solute into the cooking liquid [44]. Some authors reported that a reduced gelatinization of the starch would avoid chickpea breakage and, consequently, the migration of complex carbohydrates but not minerals and hydrophilic proteins [9,27]. Most of the proteins present in aquafaba have a molecular weight of less than 23 kDa, hence they can easily pass through the pores of the chickpea covering [27].

### 3.2. Foaming and Emulsifying Properties of the Different Types of Aquafaba

The foaming capacity and the foam stability of the different types of aquafaba and the egg white are depicted in Figure 1.

The highest foaming capacity was observed in the egg white sample with overrun values of around 180%. This foaming capacity is associated with the quantity (approximately 10–11 g/100 g) and the type of proteins present in its composition. These proteins allow the integration of numerous air bubbles during the shake and gave rise to foams with a large volume [25].

Regarding to the aquafaba samples, all of them presented a good foaming capacity with overrun values greater than 100%, although slight variations between them were observed. The results obtained in this study were similar to those described by Lafarga et al. [31], Aslan et al. [45] and Alsalman et al. [46] who reported foaming capacities between 120 and 324%.

The foaming capacity of aquafaba is related to the content of chickpeas proteins, which reduce the surface tension between the air droplets and the aqueous medium through their amphiphilic nature [47]. Stantiall et al. [25] found a very high positive correlation between the foaming capacity of aquafaba and its protein concentration; however, some discrepancies in the results obtained in our study were observed. The aquafaba samples with the lowest protein concentration (water and vegetable broth aquafaba with 1.21 and 1.19 g/100 g, respectively) were those with intermediated foaming capacity (*p* > 0.05). The canned aquafaba showed the highest foaming capacity with values practically equal (*p* > 0.05) to those of the egg white; this sample showed the highest values of protein content (2.48 g/100 g). The meat broth aquafaba was the next highest sample for its protein content (2.36 g/100 g); nevertheless, this type of aquafaba had the lowest foaming capacity. This inconsistency could be largely explained by the fat content of the sample (2.14 g/100 g). Immonen et al. [48] reported that fat in an aqueous medium hinders the formation of the foam by competing with proteins in the formation of stable films around the air droplets, interfers with the surfactant role of the proteins and reduces their foaming capacity [49]. The fat content in the meat broth aquafaba represented around the 30% of the total solids.

Relative to the foam stability, significant differences (*p* < 0.05) among the aquafaba samples and the egg white were observed. The most stability was found in the foam prepared with canned aquafaba (174% overrun), followed by the egg white (120% overrun) and the water aquafaba (116% overrun); the lowest values corresponded to the foams made with the meat and the vegetable broth aquafaba (80% and 76% overrun). Our results were similar to those described by other authors [23,31,45,50].

The vegetable and meat broth aquafaba samples showed a lower stabilizing capacity, with volume reductions (% overrun) which ranged from 37 to 44%. The low stability of the meat broth aquafaba foam was probably also related to the fat present in the dispersion. The fat competes with proteins for the adsorption at the air–water interface, resulting in the formation of interfacial films with poor viscoelastic properties and steric stability [51]. The low foam stability of vegetable broth aquafaba would be related to the lower carbohydrates content, which leads a lower foam viscosity and a higher salt concentration (vegetable broth aquafaba contains a 33% more ash content than water aquafaba), which might suppress the viscosity and the foam stability of the aquafaba [34].

The canned aquafaba showed the highest stabilizing properties and the foam maintained the same overrun percentage values as freshly prepared. This fact might be explained with a greater presence of complex polysaccharides among the total carbohydrates. The application of more intense heat treatments in the canned chickpeas than those used in the conventional pressure cooker, together with the use of less water for cooking, could accelerate the starch gelatinization as well as the solubilization and diffusion of pectins into the aquafaba, increasing the viscosity of the aqueous medium and the three-dimensional structure of the foam [52]. In addition, the higher protein concentration in the canned aquafaba, compared to the water aquafaba (2.48% versus 1.19% respectively), would facilitate the formation of more resistant films to the coalescence of the air bubbles of the foam.

The stability of the egg white foam was lower than the stability of the water and canned aquafaba foams. The foam volume remained stable or decreased by approximately 21% for canned and water aquafaba foams vs. 33% for egg white foams. This lower foaming stability could be explained by differences in the physicochemical properties between the egg and the aquafaba proteins. Soto-Madrid et al. [22] described a greater flexibility and hydrophobicity in chickpea proteins compared to egg proteins, which would contribute to create more stable films, avoiding the coalescence and separation of air bubbles. Furthermore, Mustafa et al. [23] also attributed the greater stability of the canned aquafaba foams, compared to egg white foams, to the lower content of sulfur-containing amino acids in the proteins, which better resist overwhipping by limiting the formation of disulfide bonds and the aggregation of aquafaba proteins. Finally, the pH can also play an important role in foam stability. Tufaro and Cappa observed a higher degree of syneresis in the egg white foam compared to aquafaba foam (47% vs. 27%) due to differences in their pH [33]. In our study, the pH values of egg white were approximately 8 and those of aquafaba was approximately 6, which is closer to the isoelectric point of their proteins. The approximation of the pH of aquafaba to the isoelectric point of the proteins contribute to reduce their surface charge, generating more resistant films around the air bubbles and thereby contribute to the formation of more stable foams as described by other authors [19,33]. This could be due to the higher molecular weight of globulins in plant proteins, which helps to form adsorption films with good rheological properties [19].

The emulsifying capacity and the emulsion stability of the different types of aquafaba and egg white are depicted in Figure 2.

In general, the emulsifying capacity of the different types of aquafaba were similar to those described by other authors, while the emulsion stability was much higher [24,27,53]. The egg white samples showed the highest emulsifying properties, followed by the water and vegetable broth aquafaba (*p* > 0.05); the emulsifying capacities of meat broth and canned aquafaba were significantly (*p* < 0.05) lower (around 40–50% lower than those described for egg white) [24,27,53]. The lower concentration of protein and complex polysaccharides in the aqueous phase of the water and the vegetable broth aquafaba could justify the higher emulsifying properties; in this way, the proteins could faster access the oil–water interface generated during the application of mechanical energy [54]. In addition, flexible proteins adsorbed at the interface at low concentrations are more easily denatured, favouring polymer–interface contact versus polymer–aqueous phase [54]. Moreover, the fibrous proteins present in the meat broth aquafaba and derived from the cooking of the meat are more rigid in their structure than the globular ones, so their access and adhesion to the interface would be limited and, consequently, a smaller number of emulsified fat droplets were formed. The lower emulsifying capacity of canned aquafaba could be related to the presence of complex polysaccharides in its composition. Complex polysaccharides could act, not only by hindering the adhesion of denatured proteins to the interface due to molecular entanglement, but also by competing with the proteins through the interface, preventing the emulsification of a greater proportion of oil due to the lower interfacial activity of the polysaccharides compared with proteins [54].

The emulsions prepared with water and vegetable broth aquafaba showed the lowest stability values, and the separation of the phases was observed after 120–150 min. In contrast, the emulsions prepared with meat broth and canned chickpeas aquafaba took approximately 700 to 800 min to separate; this time range was very similar to that obtained for the egg white emulsions. The gelatine present on the meat broth aquafaba and the complex polysaccharides in the case of canned aquafaba were able to retain large amounts of water, even at low concentrations, and act by increasing the viscosity or viscoelasticity of the continuous aqueous phase stabilizing the emulsion [9,55]. In addition, the higher protein content in both types of aquafaba would act to reinforce the interfaces by establishing interactions between the protein chains. Some authors, such as Bergenstahl and Claesson [56], reported a sequential mechanism for strengthening the film that covers the oil droplets derived from the interaction of the proteins attached to the interface with polysaccharides, which could help to stabilize the emulsion for longer.

### 3.3. Structure and Sensory Properties of Meringues

The alveoli percentage in cross sections of the different types of meringues made with aquafaba samples and egg white are depicted in Figure 3.

The highest alveoli degree was observed in the meringues made with egg white, followed by those prepared with vegetable broth, meat broth and water aquafaba. The lowest alveoli degree was observed in canned aquafaba meringues, which were 50% lower than those made with egg white. Our results were in accordance with those described by Mustafa et al. [23] who observed that sponge cakes made with aquafaba showed lower height and volume.

The high protein content of egg white allowed us to obtain a more homogeneous and stable foam, which was reinforced with the added sugar and could better support the baking conditions. During baking, the three-dimensional structure compacts when dehydrated and then stabilizes [39]. On the contrary, the smaller alveoli observed in the meringues made with aquafaba could be explained by the different behaviour of the chickpea proteins. Shim et al. [34] reported that the aquafaba contains more heat-stable proteins, which not be fully polymerized during the cooking of the meringues and can cause the foam to collapse.

Figure 4 shows the mean values of the three colour parameters determined in meringues: *L** (lightness), *a** (red-green index) and *b** (yellow-blue index). The results of the *L** parameter were similar in all the meringues (around 80); the lowest values were obtained in the meringues made with vegetable broth aquafaba, while the highest was with canned aquafaba. The parameters *a** and *b** showed a very similar behaviour, corresponding to the highest values for the vegetable and meat broth aquafaba meringues and the lowest for the canned aquafaba. The positive values of *a** and *b** were indicative of the predominance in the meringues of reddish and yellowish colours, respectively. These results were associated with, more or less, the development of non-enzymatic browning reactions mainly by the Maillard reaction but also by caramelization [23]. The aquafaba proteins together with large amounts of sugar and the application of temperatures of 90 °C for a prolonged time in baking cause a reduction in the water activity and a certain degree of hydrolysis of the sucrose glycosidic bonds, which increased the sugar reducing power and, consequently, the development of the Maillard reaction. The Maillard reaction was very similar in all types of meringues; the yellowish colorations stood out (*b** values of approximately +20) above the reddish ones, which were only more accentuated in the meringue samples made with vegetable broth aquafaba, which also showed a lower value in their luminosity.

Lafarga et al. [31] also described no differences in the *L**, *a** and *b** values and the chroma between fresh meringues made with aquafaba and egg white. In our study, the small differences observed in the colour parameters might be associated with the baking process, probably due to irregular heat distribution inside the oven as well as the differences in the reducing sugar content of the aquafaba. Our results were similar to those described by Mustafa et al. [23] and Nguyet et al. [28] for cupcakes, although different from those described by Stantiall et al. [25] and Tufaro and Cappa [33] for meringues made with aquafaba under similar conditions. Higher luminosity values and lower red (*a**) and yellow (*b**) indices were observed; these differences could be influenced by the type of chickpea variety and also by the cooking conditions (temperature, time, air speed or the type of oven).

The results obtained in the texture profile analysis of baked meringues prepared with the different samples of aquafaba as well as with egg white are shown in Table 2.

In all texture parameters, significant differences (*p* < 0.05) were observed between the meringues made with aquafaba and with egg white.

The fracturability values (the force necessary for the meringues collapse) were higher in the meringues made with meat broth aquafaba and showed significant differences (*p* < 0.05) with vegetable broth and canned aquafaba meringues; the water aquafaba and egg white meringues had similar values in the order of 2–3 times lower than those of the rest of the meringues. A very similar behaviour was observed in the hardness values, although in this case, all the meringues made with aquafaba showed similar results but were still significantly different (*p* < 0.05); the egg white meringues had the lowest hardness values.

The higher values of hardness and fracturability of the meringues made with aquafaba could be associated with a greater collapse of the bubbles during baking, which resulted in a greater compaction of the structure. The lower dehydration of the aquafaba proteins would contribute to this behaviour, allowing a greater association between the different protein chains, which could be reinforced by the formation of salt bonds with divalent ions released from the chickpeas during cooking. Likewise, complex polysaccharides, such as starch or pectin, and oligosaccharides solubilized during the chickpeas cooking would contribute to stabilizing the protein network of the meringue by crystallizing or forming glassy amorphous states during the cooling and causing more compact meringues, which require the application of more force to break them [47]. Finally, it should be noted that the elasticity values were practically the same and very low probably because the samples were destroyed in the first compression cycle.

Our results were very similar to those described by Stantiall et al. [25] for meringues and Aslan et al. [45] for cupcakes but differed from those described by Meurer et al. [30] and Nguyet et al. [28] who described that meringues and cupcakes made with aquafaba were less hard than those prepared with egg white.

Figure 5 shows the appearance, odour, taste and texture data of the meringues elaborated with different types of aquafaba and with egg white.

The appearance values were practically the same in all types of meringues with average scores of approximately 5. The texture values for the egg white and canned and water aquafaba meringues (scores between 5.2 and 5.7) showed significant differences (*p* < 0.05) relative to the meat and vegetable broth aquafaba meringues (scores of approximately 4.6–4.7). The greatest differences among samples were observed in the odour and taste attributes. The meringue samples prepared with meat broth and vegetable broth aquafaba had the lowest scores (approximately 4) and were significantly different (*p* < 0.05) from the other meringues. The canned aquafaba meringues were the highest scored.

These ratings were confirmed in the global impression analysis whose results are shown in Figure 6.

The meringues prepared with egg white, water and canned aquafaba obtained the highest scores (approximately 7 points out of 10), while those made with meat broth and vegetable broth aquafaba ranged between 5.8 and 6. The sapid and aromatic components from vegetables and meat contributed to the meringues aromatic scents (enhanced during the cooking) that were qualified as “strange” by the tasters, as they were not associated with what is expected for a sweet dish; therefore, these meringues received the worst scores in smell and flavour. Despite the results obtained, we consider that this aspect could be a starting point for the use of vegetable and meat broth aquafaba in the development of savoury dishes. Finally, it should be noted that the highest flavour score of the meringues made with canned aquafaba might be related with their greater compaction because of the low alveoli degree of the dough that would allow a greater contact of the meringue with the papillae during chewing.

All the meringues made with *Pedrosillano* chickpea aquafaba showed similar sensory properties to those made with egg white. Analogous results were described by Stantiall et al. [25] and Lafarga et al. [31] who found no differences in flavour, texture and the overall impression between meringues prepared with egg white and canned chickpea aquafaba.

## 4. Conclusions

The ingredients added to the cooking liquid of the chickpeas, as well as the intensity of the heat treatment during the cooking, influenced the aquafaba composition, especially the proteins, carbohydrates and fat content. Broth meat aquafaba and canned chickpeas aquafaba showed the highest concentrations for these two latter parameters.

All types of aquafaba showed good foaming properties and intermediate emulsifying capacities. The presence of proteins, complex polysaccharides and fat in the aquafaba affected the foaming and emulsifying capacities, as well as the foam and emulsion stability.

All meringues made with aquafaba displayed lower alveoli degree, greater hardness and fracturability and minimal color changes after baking than the meringues prepared with egg white. Similarly, regarding the sensory analysis, all aquafaba meringues got high scores and were similar to those obtained with egg white, except for meat and vegetable broth aquafaba, which were the lowest rated.

The results obtained in this study proved that the *Pedrosillano* chickpea aquafaba could be an egg substitute in the development of different cooking recipes (sweet or savory), which involved foams or emulsions and could be an alternative for consumers with an egg allergy. The aquafaba derived from commercial canned chickpeas showed, in general, more similarities to the properties of egg white and had the highest sensory scores from the meringues prepared with it.

## Figures and Tables

**Figure 1 foods-12-00902-f001:**
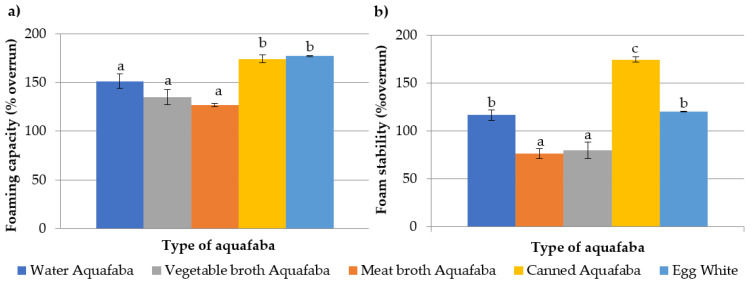
Foaming capacity (**a**) and foam stability (**b**) expressed in % of overrun of *Pedrosillano* chickpea aquafaba obtained with different broths and egg white. ^a–c^ Means with different superscripts showed significant differences (*p* < 0.05).

**Figure 2 foods-12-00902-f002:**
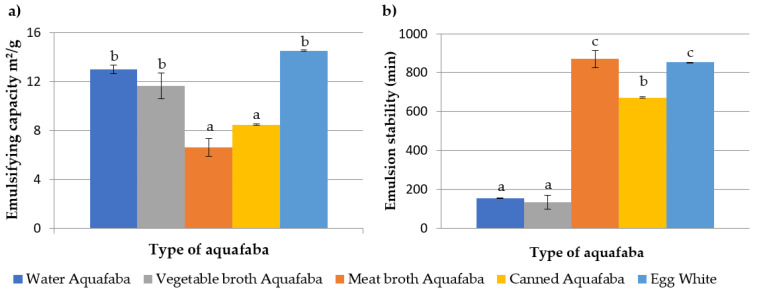
Emulsifying capacity, expressed as m^2^/g (**a**), and emulsion stability, expressed in min (**b**), of *Pedrosillano* chickpea quafaba obtained with different broths and egg white. ^a–c^ Means with different superscripts showed significant differences (*p* < 0.05).

**Figure 3 foods-12-00902-f003:**
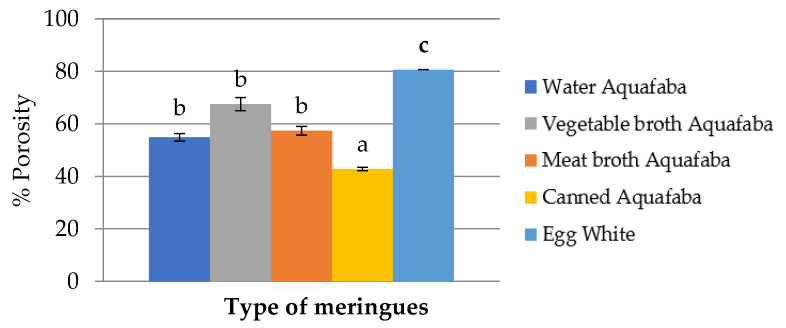
Mean values of alveoli degree of meringues, expressed as % of porosity, made with four types of *Pedrosillano* chickpea aquafaba and egg white. ^a–c^ Means with different superscripts showed significant differences (*p* < 0.05).

**Figure 4 foods-12-00902-f004:**
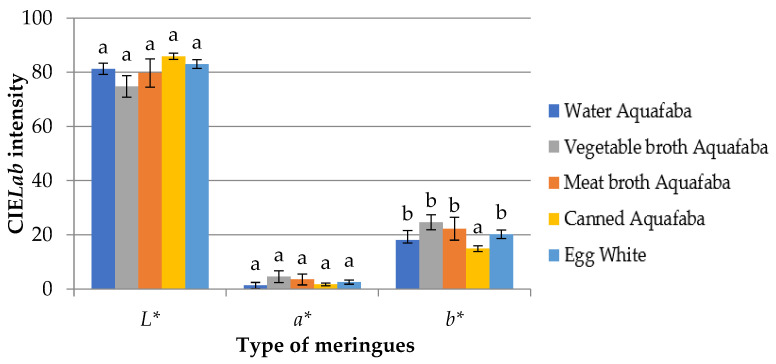
Mean values of CIE*Lab* parameters in meringues made with four types of *Pedrosillano* chickpea aquafaba and egg white. Luminosity (*L**), red-green (*a**), yellow-blue (*b**). ^a,b^ Means for each CIE*Lab* parameter with different superscripts showed significant differences (*p* < 0.05).

**Figure 5 foods-12-00902-f005:**
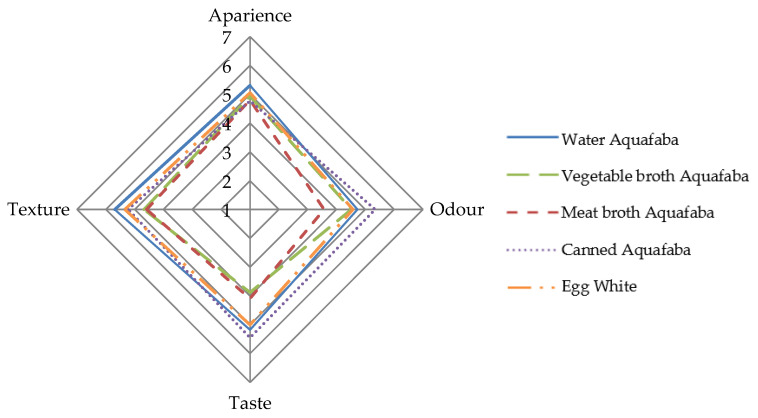
Average scores obtained during the sensory analysis of the different types of meringues made with *Pedrosillano* chickpea aquafaba and egg white using a 7-points hedonic scale (1 = dislike extremely; 7 = like very much).

**Figure 6 foods-12-00902-f006:**
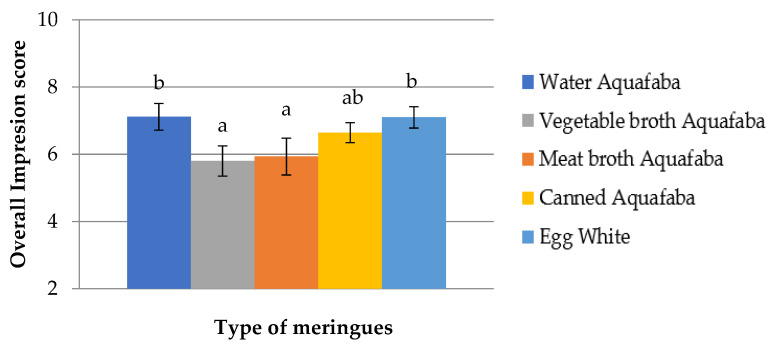
Overall impression scores of meringues made with *Pedrosillano* chickpea aquafaba and egg white using a rating from 1 (very bad) to 10 (very good). ^ab^ Means with different superscripts showed significant differences (*p* < 0.05).

**Table 1 foods-12-00902-t001:** Mean values (expressed as g/100 g aquafaba) and standard deviation of the chemical composition of different types of chickpeas aquafaba.

	Total Solids (%)	Protein (%)	Carbohydrates (%)	Ash (%)	Fat (%)
**Water Aquafaba**	5.84 ± 0.08 ^a^	1.21 ± 0.01 ^a^	3.79 ± 0.08 ^c^	0.77 ± 0.02 ^a^	nd
**Vegetable broth Aquafaba**	5.63 ± 0.15 ^a^	1.19 ± 0.04 ^a^	3.36 ± 0.13 ^c^	1.03 ± 0.08 ^b^	nd
**Meat broth Aquafaba**	7.84 ± 0.11 ^c^	2.36 ± 0.05 ^b^	2.20 ± 0.11 ^b^	1.12 ± 0.01 ^b^	2.14 ± 0.23 ^b^
**Canned Aquafaba**	6.24 ± 0.05 ^b^	2.48 ± 0.03 ^b^	1.46 ± 0.09 ^a^	1.15 ± 0.03 ^b^	1.15 ± 0.72 ^a^

^a–c^ Means in the same column with different superscripts showed significant differences (*p* < 0.05); nd: not detected.

**Table 2 foods-12-00902-t002:** Mean values and standard deviation obtained in the texture profile analysis of meringues made with four types of *Pedrosillano* chickpea aquafaba and egg white.

	Water Aquafaba	Vegetable Broth Aquafaba	Meat Broth Aquafaba	Canned Aquafaba	Egg White
**Fracturability (N)**	3.76 ± 0.41 ^a^	9.02 ± 2.71 ^c^	12.73 ± 2.44 ^d^	9.26 ± 2.79 ^c^	5.23 ± 1.09 ^b^
**Hardness (N)**	39.33 ± 4.89 ^b^	49.18 ± 2.72 ^c^	56.17 ± 4.83 ^d^	45.28 ± 2.28 ^c^	14.34 ± 1.51 ^a^
**Springiness**	0.05 ± 0.01 ^a^	0.08 ± 0.02 ^ab^	0.17 ± 0.04 ^b^	0.07 ± 0.01 ^ab^	0.05 ± 0.02 ^a^

^a–d^ Means in the same row with different superscripts showed significant differences (*p* < 0.05). N = Newtons.

## Data Availability

Data is contained within the article.

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
