# Peer review of "Study of the Technological Properties of Pedrosillano Chickpea Aquafaba and Its Application in the Production of Egg-Free Baked Meringues"

_foods, 2023, doi:10.3390/foods12040902_

Round 1

Reviewer 1 Report

The manuscript is well prepared and explains the immportance of aquafaba products in egg replacement for the food industry.

I would have liked to see triplicate analysis conducted rather than duplicate (especially as some of the error bars are large), however I understand that this will not be possible to complete.

The paper is written well and generally all the information presented appropriately.

Figure 1 (B) is interesting in relation to the statistics and the error bars. Could the authors just check their data once more and ensure all is correct?

Could the authors also explain in a little more detail about the differences in foam stability and give examples of similar research to illustrate how their results compare with those of others ?

With the results presented in Table 2 there seems to be an error made in the calulation of significant differences. Could you please recheck your results and ensure that statistical differences are reported accurately ?

With regards to the references, two new papers have come out in the last year which you may consider using in your paper

Development of innovative clean label emulsions stabilized by vegetable proteins

Marta CabritaSara SimõesEstefanía Álvarez-CastilloDiogo Castelo-BrancoAna TassoDiogo FigueiraAntonio GuerreroAnabela Raymundo International Journal of Food Science & TechnologyVolume 58, Issue 1    

Effects of concentration methods on the characteristics of spray-dried black soybean cooking water

Esteban Echeverria-JaramilloWeon-Sun Shin International Journal of Food Science & TechnologyVolume 57, Issue 11

First published: 17 September 2022

Author Response

REPONSE TO REVIEWER 1 COMMENTS

Dear reviewer,

We would like to express our gratitude for the constructive remarks and helpful suggestions that have undoubtedly improved our manuscript. We took your recommendations into full consideration, please find the answers to all your remarks below.

R1: Figure 1 (B) is interesting in relation to the statistics and the error bars. Could the authors just check their data once more and ensure all is correct?

A: Thank you so much for your annotation. The authors revised the statistical analysis of the all data and some mistakes were detected. The corrected results and changes in Figure 1 were included in the new version of the manuscript.

R1: Could the authors also explain in a little more detail about the differences in foam stability and give examples of similar research to illustrate how their results compare with those of others?

A: Thank you so much for your comment. Changes in the revised version of the manuscript were incorporated.

R1: With the results presented in Table 2 there seems to be an error made in the calculation of significant differences. Could you please recheck your results and ensure that statistical differences are reported accurately?

A: Thank you so much for your valuable comment. The authors revised the statistics and some mistakes were detected. The accurate results were included in the new version of the manuscript.

R1: With regards to the references, two new papers have come out in the last year which you may consider using in your paper

Development of innovative clean label emulsions stabilized by vegetable proteins. Marta Cabrita, Sara Simões, Estefanía Álvarez-Castillo, Diogo Castelo-Branco, Ana Tasso, Diogo Figueira, Antonio Guerrero, Anabela Raymundo. International Journal of Food Science & TechnologyVolume 58, Issue 1.

Effects of concentration methods on the characteristics of spray-dried black soybean cooking water. Esteban Echeverria-Jaramillo, Weon-Sun Shin International Journal of Food Science & TechnologyVolume 57, Issue 11

A: Thank you so much for your recommendation. Both references were included in the revised version of the manuscript. Please, see in the accompanying file.

Reviewer 2 Report

The authors evaluated certain interfacial properties of chickpea aquafaba prepared in different ways and compared results to egg whites. Moreover, merengues were prepared from these liquids, of which a sensory analysis conducted. The overall objective seems to have been to achieve different flavor profiles of products with aquafaba by using vegetable or meat broth as cooking media. This may be of interest for applications in a gastronomy or maybe even industrial setting. However, extensive revisions are needed. There are many grammatical and stylistic errors throughout the manuscript and the discussion should be more in-depth. Below some more comments:

L25: I was initially very confused about the term "meat aquafaba". After reading the methods, it became clear. But please modify the description of the study design in the abstract.

L63: In the interest of a balanced compilation of literature, the authors should also mention that some studies have been critical towards aquafaba as egg substitute and stated that based on their analysis, it is not superior to egg in its environmental impact. See for example Saget et al. (2021) Sustainability 13 (9), 4726.

L81 & also in other places (e.g., L57). As shown in Table 1, 100 g of aquafaba contain less than 2.5 g protein. How is that "protein-rich"? I agree with the authors that aquafaba is an interesting culinary ingredient bu such claims overstate the nutritional benefits and sound more like marketing than a scientific text. Please carefully check your manuscript (not only in L81!) and make sure not to exaggerate (e.g., "intense and balanced flavor")! 

L82: The statement about denaturation demands some elaboration and references. I agree with the authors that usually a partial denaturation step improves interfacial properties of proteins, but has this been shown for chickpea proteins? If so, include a reference and be more specific about what has been found. If not, then include a reference where it has been shown for a different protein type and explain that this is common for proteins and therefore can be expected for aquafaba.

L87: Would these be intrinsic carbohydrates or additives (such as gums)?

L94: I disagree with the statement "practically no studies on its chemical composition". The authors have in fact cited multiple studies that reported on the chemical composition. I agree that there still information lacking, and several studies did not go very deep in their analysis, but it is not true that there are "practically no studies". Also, since there is work on sponge cakes, "practically no" is not the right term (there is limited work, but it exists).

L109: The authors refer to it as vegetable aquafaba, but is was vegetable broth. So maybe calling the samples meat broth and vegetable broth aquafaba would be clearer.

L115: Using meat broth was a surprising choice. Aquafaba is popular in the vegan community, would preparing it in meat broth not severely limit the targeted consumer group? Moreover, the environmental benefits are also eliminated this way.

L143: "indications" is not a good word choice, and what is "the separatory funnel method"? It sounds odd

L144: "calcination" - - -> (dry) ashing?

L157 - 169: The authors should explain what the variables in the equation mean. Also, absorbance is typically abbreviated as A not Abs

L172: Could the authors also use words other than "elaborated", such as "prepared" or "made"? To me, "elaborated" sounds a bit strange, but that may be because of regional use of the term.

L200: Isn't "alveolate" an adjective. So grammatically, it does't really work in this sentence.

There are some formatting issues for Table 1, the line numbers overlap with text.

L250 & 256 & 395: "presented" does not sound like the best word choice here. How about "contained"?

L261: This sentence is not grammatically correct.

L284: external covering? Do the authors mean seed coat?

L286: "would not have caused excessive gelatinization" - the authors do not have data on this. 

L308: Typically when letters are assigned to denote statistical differences, the values are ranked and whatever is the highest gets an 'a', the next-highest a 'b', and so on. Switching the order and assigning an 'a' to the lowest, 'b' to second-lowest etc. is also possible. But the authors are doing neither in Fig. 1b and 2b.

L317: Please remove "a good" here, because it is a subjective assessment

L321: subject-verb disagreement

L321-323: I think the authors are referring to previous studies here, not their own results. However, the sentence does not make that clear and should thus be modified.

L324: "verified" does not seem the right word choice

L327: Remove the d at the end of intermediated. The last part of the sentence does not sound grammatically correct (in the context of this sentence).

L349 & 381: "justified" is not a good word choice; also, the authors do not have data on the carbohydrate composition. Total carbohydrates were determined by difference (the same comment can be made about the sentence in L380). The authors also did not measure viscosity so the next paragraph is not supported by their own data. Throughout the manuscript, the authors need to make it clear which statements refer to observations by other authors.

L375: Replace "best" by "highest"

L387: If the meat proteins were less soluble, then how did they end up in aquafaba? If they were actually particles, then shouldn't the aquafaba have been filtered?

There is another formatting issue with the y-axis and line numbers overlapping for table 3. Same for figures 4 and 6.

L432: When egg white proteins set the structure, they polymerize. Is that not what causes solidification of the foams? That is different than denaturation.

L440 and below: Typically, L, a and b are italiziced. 

L506: Again, the authors did not assess the extent of denaturation so this statement is based on literature and not their own data

L512: Would they really crystallize or form glasses during baking? Are these not processes that happen during cooling?

L556 & elsewhere: replace "best" by "highest"

L580: Remove "excellent"

L582: Is there evidence for this statement or is this a hypothesis?

L594: "both parameters" - which ones? In the previous sentence, the authors are listing proteins, carbohydrates and fat.

Author Response

REPONSE TO REVIEWER 2 COMMENTS

Dear reviewer,

We would like to express our gratitude for the constructive remarks and helpful suggestions that have undoubtedly improved our manuscript. We took your recommendations into full consideration, please find the answers to all your remarks below.

R2: L25: I was initially very confused about the term "meat aquafaba". After reading the methods, it became clear. But please modify the description of the study design in the abstract.

A: Thank you so much for your remark. The abstract was modified in the revised version of manuscript: “The aim of this study was to assess the compositional differences and the culinary properties of Pedrosillano chickpea aquafaba prepared with different cooking liquids (water, vegetable broth, meat broth and the covering liquid of canned chickpeas), and to evaluate the sensory characteristics of French baked meringues made with the different aquafaba samples using egg white as control”.

R2: L63: In the interest of a balanced compilation of literature, the authors should also mention that some studies have been critical towards aquafaba as egg substitute and stated that based on their analysis, it is not superior to egg in its environmental impact. See for example Saget et al. (2021) Sustainability 13 (9), 4726.

A: Thank you so much for your comment. This information was included in the corrected manuscript “However, it is necessary to consider that a recent study [13] questioned whether the cultivation of legumes had less environmental impact than eggs production”

R2: L81 & also in other places (e.g., L57). As shown in Table 1, 100 g of aquafaba contain less than 2.5 g protein. How is that "protein-rich"? I agree with the authors that aquafaba is an interesting culinary ingredient but such claims overstate the nutritional benefits and sound more like marketing than a scientific text. Please carefully check your manuscript (not only in L81!) and make sure not to exaggerate (e.g., "intense and balanced flavor")!

A: Thank you for your comment. The manuscript was revised, and those claims related to nutritional and/or sensory properties that could seem exaggerated or subjective were removed from the text.

For example:

  • Lines 55-56: The presence of some proteins, complex carbohydrates and other flavour components turns aquafaba into an ingredient of exceptional culinary quality due to
  • Lines 71-72: … of a growing number of consumers, who prefer dietary ingredients rich in fibre and/or allergen-free
  • Lines 80-81: In the case of aquafaba, the foam production is also possible due to the presence of the chickpea proteins.
  • Lines 597-599: The results obtained in this study suggest that the Pedrosillano chickpea aquafaba could be an egg substitute in the development of different cooking recipes (sweet or savory) which involved…

R2: L82: The statement about denaturation demands some elaboration and references. I agree with the authors that usually a partial denaturation step improves interfacial properties of proteins, but has this been shown for chickpea proteins? If so, include a reference and be more specific about what has been found. If not, then include a reference where it has been shown for a different protein type and explain that this is common for proteins and therefore can be expected for aquafaba.

A: Following the reviewer´s recommendation, the references of two studies that justify the statement pointed out by the reviewer were included in the line 83 of the revised manuscript:

  • Soto-Madrid, D.; Perez, N.; Gutierrez-Cutino, M.; Matiacevich, S.; Zúñiga, R.N. Structural and Physicochemical Characterization of Extracted Proteins Fractions from Chickpea (Cicer arietinum L.) as a Potential Food Ingredient to Replace Ovalbumin in Foams and Emulsions. Polymers 2023, 15, 110.
  • Zhi-gang HuangXue-ying WangJia-yi ZhangYi LiuTong ZhouShang-yi ChiChong-hao Bi. High-pressure homogenization modified chickpea protein: Rheological properties, thermal properties and microstructure. Journal of Food Engineering 335 (2022) 111196.

R2: L87: Would these be intrinsic carbohydrates or additives (such as gums)?

A: The authors refer to the intrinsic polysaccharides of the chickpea, this clarification was included in the manuscript

R2: L94: I disagree with the statement "practically no studies on its chemical composition". The authors have in fact cited multiple studies that reported on the chemical composition. I agree that there still information lacking, and several studies did not go very deep in their analysis, but it is not true that there are "practically no studies". Also, since there is work on sponge cakes, "practically no" is not the right term (there is limited work, but it exists).

A: The authors meant there are few studies that included in the same work the analysis of the chemical composition of different aquafaba samples and their subsequent use in the preparation of meringues. Nevertheless, we consider the reviewer's comments is correct and “practically no studies” was replaced by “limited studies” in the revised version of the manuscript.

R2: L109: The authors refer to it as vegetable aquafaba, but is was vegetable broth. So maybe calling the samples meat broth and vegetable broth aquafaba would be clearer.

A: Thank you for your comment. The authors agree with the reviewer opinion, and changes in the manuscript were accomplished.

R2: L115: Using meat broth was a surprising choice. Aquafaba is popular in the vegan community, would preparing it in meat broth not severely limit the targeted consumer group? Moreover, the environmental benefits are also eliminated this way.

A: The authors understand the reviewer's reflection. However, in this work we did not want to limit to the use of vegetable broths, but also to include other types of broths that are very common in traditional and avant-garde cuisine in many countries around the world. The authors think that the study of other culinary alternatives would contribute to expanding the knowledge about aquafaba, opening up new possibilities for chefs or companies in the development of dishes or products intended for the population that is not vegan or vegetarian, or that may have problems with egg allergies.

R2: L143: "indications" is not a good word choice, and what is "the separatory funnel method"? It sounds odd

A: Thank you for your comments. These aspects were corrected in the revised version of the manuscript: “The fat content was quantified by AOAC standard 920.39 using a Soxhlet extraction system [34]”

R2: L144: "calcination" - - -> (dry) ashing?

A:Calcination” was replaced by (dry) ashing in the revised manuscript: “The ash content was determined by gravimetry, using a dry ashing method with a muffle at 500°C for 6 h”

R2: L157 - 169: The authors should explain what the variables in the equation mean. Also, absorbance is typically abbreviated as A not Abs

A: Thank you for your remark. The meaning of each variable was included in the revised manuscript, and “Abs” was changed by “A”.

R2: L172: Could the authors also use words other than "elaborated", such as "prepared" or "made"? To me, "elaborated" sounds a bit strange, but that may be because of regional use of the term.

A: Thank you for your suggestion, we agree with the reviewer; the word “elaborated” was changed by “prepared” or “made” in the revised manuscript

R2: L200: Isn't "alveolate" an adjective. So grammatically, it does't really work in this sentence.

      There are some formatting issues for Table 1, the line numbers overlap with text.

A: The sentence was modified “The percentage of the meringues area that corresponded to the alveoli was determined by taking cross section photographs, which were subsequently divided into portions of 1 cm long”. Likewise, Table 1 was revised to avoid the overlapping with the line numbers.

R2: L250 & 256 & 395: "presented" does not sound like the best word choice here. How about "contained"?

A: Thank you for your suggestion, we agree the reviewer, modifications were carried out in the revised version of the manuscript.

R2: L261: This sentence is not grammatically correct.

A: This sentence was corrected in the revised manuscript.

R2: L284: external covering? Do the authors mean seed coat?

A: Thank you for your comment, we agree with the reviewer; the expression “external covering” was changed by “seed coat” in the revised manuscript

R2: L286: "would not have caused excessive gelatinization" - the authors do not have data on this.

A: The authors agree with the reviewer's observation. The authors tried to explain in a general way the importance of gelatinization in the passage of solutes from the chickpea to the aquafaba during the cooking. In this sense, this sentence was rewritten in the revised manuscript.

R2: L308: Typically when letters are assigned to denote statistical differences, the values are ranked and whatever is the highest gets an 'a', the next-highest a 'b', and so on. Switching the order and assigning an 'a' to the lowest, 'b' to second-lowest etc. is also possible. But the authors are doing neither in Fig. 1b and 2b.

A: The letters assigned to denote statistical differences were corrected in the figures and the tables of the revised manuscript in the way suggested by the reviewer (“a” to the lowest value, “b” to second-lowest and so on).

R2: L317: Please remove "a good" here, because it is a subjective assessment

A: Thank you for your comment The term “a good” was removed in the manuscript.

R2: L321: subject-verb disagreement

A: This sentence was corrected in the revised manuscript.

R2: L321-323: I think the authors are referring to previous studies here, not their own results. However, the sentence does not make that clear and should thus be modified.

A: The reviewer is right. These sentences were changed in the manuscript: “Similarly to egg white, the foaming capacity of aquafaba is related to the quantity of proteins, which act by reducing the surface tension between the air droplets and the aqueous medium because of its amphiphilic nature [39].”

R2: L324: "verified" does not seem the right word choice

A: Thank you for your comment, “verified” was changed by “found” in the revised version of the manuscript

R2: L327: Remove the d at the end of intermediated. The last part of the sentence does not sound grammatically correct (in the context of this sentence).

A: The letter d was removed from “intermediated” and the sentence was corrected in the revised manuscript

R2: L349 & 381: "justified" is not a good word choice; also, the authors do not have data on the carbohydrate composition. Total carbohydrates were determined by difference (the same comment can be made about the sentence in L380). The authors also did not measure viscosity so the next paragraph is not supported by their own data. Throughout the manuscript, the authors need to make it clear which statements refer to observations by other authors.

A: “justified” was changed by “explain” in the manuscript. In addition, the authors took in consideration the reviewer´s comment, and all the manuscript was revised to make more clear what statements were referred to the observations of other authors.

R2: L375: Replace "best" by "highest"

A: The replacement indicated by the reviewer was carried out in the manuscript

R2: L387: If the meat proteins were less soluble, then how did they end up in aquafaba? If they were actually particles, then shouldn't the aquafaba have been filtered?

A: “less soluble” was removed from the text. All the aquafaba samples were filtered.

R2: There is another formatting issue with the y-axis and line numbers overlapping for table 3. Same for figures 4 and 6.

A: The formatting problems in table 3 and in figures 4 and 6 were corrected

R2: L432: When egg white proteins set the structure, they polymerize. Is that not what causes solidification of the foams? That is different than denaturation.

A: The reviewer is right. This aspect was corrected in the manuscript. “…which would not be fully polymerized during the cooking of the meringues and caused the foam to collapse”

R2: L440 and below: Typically, L, a and b are italiziced.

A: Thank you, letters L, a and b were italiziced in the text.

R2: L506: Again, the authors did not assess the extent of denaturation so this statement is based on literature and not their own data

A: Thank you for your comment, changes in the revised version of the manuscript were carried out. The authors reference to denaturation was removed. “The dehydration of aquafaba proteins would contribute to this behaviour, allowing a greater association between…….”

R2: L512: Would they really crystallize or form glasses during baking? Are these not processes that happen during cooling?

A: Thank you for your annotation. The reviewer is right. Crystallization or the formation of glassy amorphous states occurs during the cooling of meringues. These aspects have been corrected in the revised manuscript: “…would contribute to stabilizing the protein network of the meringue by crystallizing or forming glassy amorphous states during cooling…

R2: L556 & elsewhere: replace "best" by "highest"

R2: L580: Remove "excellent"

A: Thank you for your comments, all the suggestions were included in the manuscript.

R2: L582: Is there evidence for this statement or is this a hypothesis?

A: It is a hypothesis. This sentence was rewritten. “Finally, it should be noted that the highest flavour score of the meringues made with canned aquafaba might be related with the greater compaction because of the low degree of alveolate of the dough, that would what allow a greater contact of the meringue with the papillae during chewing.”

R2: L594: "both parameters" - which ones? In the previous sentence, the authors are listing proteins, carbohydrates and fat.

A: Thank you for your comment. The sentence was modified in the manuscript: “The ingredients added to the cooking liquid of the chickpeas, as well as the intensity of the heat treatment during the cooking, influenced in the aquafaba composition, especially on proteins, carbohydrates and fat contents. Broth meat and canned aquafaba showed the highest concentrations for these two latter parameters”.

Reviewer 3 Report

In the current research, the compositional culinary properties of aquafaba were evaluated which is a by-product derived from the legumes processing. The innovative and scientific of the research is not clear. It is failed to put forward a scientific problem or theoretical hypothesis. The experimental design scheme and technical means are common. The composition of four samples (Water Aquafaba, Vegetable Aquafaba, Meat Aquafaba, Canned Aquafaba) were too complex to be used as control experimental groups. Other components in the experimental group may interfere culinary properties of aquafaba. Therefore, the study suggested that the aquafaba could be an exceptional egg substitute is not appropriate. The experimental methods and technical means are common. The physical factors (whipping, microwave, pressure, heat treatment, etc.) and chemical factors (salt, pH, sugar, etc.) should be considered. Foaming and emulsifying properties can also be characterized by other indicators, such us interfacial property, rheological property, etc. The abstract should be revised. It should be clear and concise. There are too many keywords that should be decreased to 5 words.

Author Response

REPONSE TO REVIEWER 3 COMMENTS

R·3: In the current research, the compositional culinary properties of aquafaba were evaluated which is a by-product derived from the legumes processing. The innovative and scientific of the research is not clear. It is failed to put forward a scientific problem or theoretical hypothesis. The experimental design scheme and technical means are common. The composition of four samples (Water Aquafaba, Vegetable Aquafaba, Meat Aquafaba, Canned Aquafaba) were too complex to be used as control experimental groups. Other components in the experimental group may interfere culinary properties of aquafaba. Therefore, the study suggested that the aquafaba could be an exceptional egg substitute is not appropriate. The experimental methods and technical means are common. The physical factors (whipping, microwave, pressure, heat treatment, etc.) and chemical factors (salt, pH, sugar, etc.) should be considered. Foaming and emulsifying properties can also be characterized by other indicators, such us interfacial property, rheological property, etc. The abstract should be revised. It should be clear and concise. There are too many keywords that should be decreased to 5 words.

A: Dear reviewer, we would like to express our gratitude for your comments.

In general, the studies on aquafaba have been based on the use of aquafaba derived from commercial canned legumes, and to the best of the authors' knowledge, the study of the chemical characteristics and functional properties of the aquafaba obtained during the preparation of dishes has not been addressed. From the scientific point of view, the starting hypothesis of this research work was: “Could the cooking liquid of Pedrosillano chickpeas under different culinary recipes widely established in gastronomy be a good substitute for eggs as an ingredient in the manufacture of sweet or salty baked meringues?” Our work also contributes a technological application such as the development and innovation in obtaining egg-free sweet or salty meringues. This aspect has been included in the revised manuscript.

The authors agree that the composition of the different aquafaba samples studied are complex, but disagree that they cannot be used as control. To carry out each experiment, the most commonly conditions applied into the kitchen were used, standardizing the weights of the raw materials, times and cooking equipment used in the preparation of each of the aquafaba sample. In addition, all the processes for preparing the broths and cooking the chickpeas were carried out at least in duplicate. The results of this study provide scientific and technological data that conclude that aquafaba can be an appropriate ingredient (the authors agree with the reviewer that the term “exceptional” is not adequate) as an egg substitute for the development of traditional and modern dishes.

In addition, the authors wish to state that this research paper is a preliminary study and that most of the physical and chemical factors noted by the reviewer were taken into account. Thus, the beating conditions, heat treatment, cooking time, pressure, pH of the aquafaba, amount of sugar used in the manufacture of meringues have been the same in all the experiments. No microwave cooking treatment was used in this study.

Moreover, the authors agree with the reviewer on the different possibilities that could be used to analyse foaming and emulsifying properties. However, in this work we proceeded to use techniques widely approved and used by the scientific community for the study of these functional properties. It should be noted that the objective of the work was not only focused on the study of the surface properties of aquafaba, but also on its subsequent technological application.

In addition, the abstract was revised and some modifications were included. In the same way, some of the keywords of the original manuscript were removed.

Finally, the authors would thank the reviewer for his comments, as they will undoubtedly contribute to improve the experimental design and the methodology to be used in future research work related to the topic of this article.

Round 2

Reviewer 2 Report

The authors have for the most part addressed reviewer comments sufficiently.

Some minor comments:

The authors use the term legumes, but should it not be pulses? In L37, the authors write that legumes are edible seeds from the Leguminoseae family, but doesn't the term legumes refer to the plant while the seeds are called pulses?

In L29, I think the word "panel-tester" should be replaced by "sensory panelist".

In the abstract, the article can be removed in L11 and L22 before the words "legumes" and "aquafaba", respectively. 

In L58, I would replace "the legumes processed" with "legume processing"

L93 - 96: The sentence is a bit strange in the way it is worded, probably because of the changes made. The grammar is off.

L115: Was this a hypothesis or was the aim to evaluate interfacial, textural and sensory properties of aquafaba prepared by using different cooking media? A hypothesis can be decided on a yes/no basis, due to statistics. But "good substitute" is quite vague.

L341: I thought the term chickpea covering was going to be replaced by seed coat?

There is still a formatting issue with the figures, the y-axis labels overlap with line numbers.

Author Response

REPONSE TO REVIEWER 2 COMMENTS

Dear reviewer,

Once again, we would like to express our gratitude for your constructive remarks. We took your recommendations into full consideration, please find the answers to all your remarks below

 R2: The authors use the term legumes, but should it not be pulses? In L37, the authors write that legumes are edible seeds from the Leguminoseae family, but doesn't the term legumes refer to the plant while the seeds are called pulses??

A: Thank you so much for your annotation. The reviewer is right, “legumes” was changed by pulses in the final version of the manuscript.

 R2: In L29, I think the word "panel-tester" should be replaced by "sensory panelist".

A: Thank you so much for your comment. Changes in the revised version of the manuscript were incorporated (“panel-tester” was changed by "sensory panelist").

 R2: In the abstract, the article can be removed in L11 and L22 before the words "legumes" and "aquafaba", respectively.

A: Thank you, both articles were deleted in the manuscript.

 R2: In L58, I would replace "the legumes processed" with "legume processing"

A: Thank you, the replacement suggested by the reviewer was accomplished in the new version of the manuscript.

R2: L93 - 96: The sentence is a bit strange in the way it is worded, probably because of the changes made. The grammar is off.

A: Thank you so much for your valuable comment, the reviewer is right. The authors revised and rewritten the paragraph in the manuscript “In the case of aquafaba, the foam production is also possible due to the presence of an enough quantity of chickpeas proteins dissolved in the cooking broth. Those proteins require a partial denaturation, so that the hydrophobic amino acids were oriented towards the air bubbles and the hydrophilic ones towards the aqueous phase favouring the formation and the maintenance of the structure [21,22,23].

R2: L115: Was this a hypothesis or was the aim to evaluate interfacial, textural and sensory properties of aquafaba prepared by using different cooking media? A hypothesis can be decided on a yes/no basis, due to statistics. But "good substitute" is quite vague.

A: Thank you for your annotation. The authors agree with the reviewer. The hypothesis was changed in the manuscript “Can the cooking liquid of Pedrosillano chickpeas, under different culinary recipes widely established in gastronomy, present similar functional properties and sensory characteristic to the egg white as an ingredient in the manufacture of sweet or salty baked meringues?”

R2: L341: I thought the term chickpea covering was going to be replaced by seed coat?

A: Sorry about the author´s mistake; “chickpea covering” was changed by “seed coat”.

R2: There is still a formatting issue with the figures, the y-axis labels overlap with line numbers.

A: Sorry about this problem. In the author´s document this overlapping did not exist. The authors think that in the final document will have no overlapping problems since the number lines will not appear in the final PDF document.

Reviewer 3 Report

The manuscript has been revised according to the reviewer comments. And it has been sufficiently improved compared with the previous version. I think it is suitable to publish in Foods.

Author Response

REPONSE TO REVIEWER 3 COMMENTS

Dear reviewer,

Once again, we would like to express our gratitude for your comments that have undoubtedly improved our manuscript.

Thank you so much.